# The Effect of Green Human Resources Management Practices on Employees' Affective Commitment and Work Engagement: The Moderating Role of Employees' Biospheric Value

Jorge F. S. Gomes [1], Ana Sabino [2,3,*] and Vanessa Antunes [4]

1   Advance/CSG, ISEG, School of Economics and Management, University of Lisbon, Rua do Quelhas 6, 1200-781 Lisbon, Portugal
2   School of Psychology, ISPA—Instituto Universitário, R. Jardim do Tabaco 34, 1149-041 Lisbon, Portugal
3   APPsyCI—Applied Psychology Research Center Capabilities and Inclusion, ISPA Instituto Universitário, 1149-041 Lisboa, Portugal
4   Human Resource Specialist, AskBlue, 1700-239 Lisboa, Portugal
*   Correspondence: asabino@ispa.pt

**Abstract:** Organizational Sustainability is an increasingly important movement in the business world because of its social impact and also for the obligations imposed by state agendas and programs by global entities, such as the United Nations. At the forefront of such a movement is the Human Resources function, given its boundary activities with several critical internal and external stakeholders. The term Green Human Resource Management (GHRM) has been used to describe people management with a focus on green issues. The main objective of this research was to explore the relationship between personal values associated with sustainable behaviors (altruistic, biospheric, egoistic, and hedonic values), organizational outcomes such as engagement and commitment, and GHRM. A questionnaire was prepared and used to collect 532 responses from employees from various organizations. Results show that of the four personal values only the biospheric one moderates the relationship between GHRM practices and affective commitment so that the relationship between HRM and affective commitment is stronger when biospheric values score higher. This is an important finding, as it shows that when people value the biosphere, the effect of GHRM practices on affective commitment and work engagement is stronger than when people value other matters.

**Keywords:** green human resources management; affective commitment; work engagement; personal environmental values; hedonic value; biospheric value; egoistic value; altruistic value

## 1. Introduction

The future is shaped not only by economic and financial megatrends but also by the short- and long-term actions and decisions of countries, industries, organizations, and individuals. This view of a deeply interconnected and interdependent world is triggering an increasing interest in sustainability. More than a new buzzword, sustainability is a must, as it highlights the relevance of daily choices aimed at avoiding ecological collapse and ensuring that upcoming generations cohabit in a healthy environment and develop an appreciation for human integrity [1]. Both at national and international levels, organizations are compelled to adopt measures to control their ecological footprint and assume a sustainable role in their market performance. Sustainability is increasingly found in international political agendas and at the center of many debates led by the United Nations [2,3].

Outlined in 2015 by the UN, the 17 Sustainable Development Goals (SDG) were presented and implemented through the European Union's 2030 Agenda and are grouped into six major themes: dignity, people, planet, partnerships, justice, and prosperity. They are framed by the Triple Bottom Line (TBL) approach, which comprises three types of sustainability: (i) economic sustainability, related to liquidity and profit; (ii) social sustainability,

focused on people and society and promoting an ethical concern; and (iii) environmental sustainability, which highlights the planet and its resources [2,4–6]. Considering the environmental catastrophes that the world is increasingly witnessing, the present work focuses on the environmental pillar for its specificity and relevance. This pillar intends to establish limits and outline a short- and long-term environmental plan suitable for the entire global society [7]. It is also important to note that both TBL theory and the SDG were defined to be applied by countries, organizations, and individuals [8], guaranteeing an increasing responsibility for all players to ensure sustainable development.

Organizations have been raising their awareness and sense of responsibility for environmental sustainability, reflected in different sectors, from Industry 4.0 and manufacturing [9], to tourism [10], and to marketing [11]. Given its direct influence on employees, the human resource management (HRM) function seems to be at the center of the debate [12–14]. In fact, the HRM function assumes a critical role in motivating employees to support environmental concerns, through the development of sustainable practices; this has come to be known as green HRM [15]. In a recent review of GHRM [16], Ren and colleagues proposed a conceptual framework that highlighted organizational and individual outcomes along two dimensions. The first one, called green outcomes, comprises organizational outcomes, and ranges from environmental performance and environmental innovation, to individual outcomes such as green citizenship behavior and green competencies. The second dimension is related to non-green outcomes, and includes financial performance, reputation, and turnover. The present study addresses two non-green and individual outcomes of GHRM practices: affective commitment and work engagement.

These two constructs are fundamental in organizational behavior studies and highly related to competitiveness and performance [17]. Organizational commitment is defined as the emotional bond between the employee and the organization [18], and work engagement is a fulfilling state of mind characterized by high levels of vigor, dedication, and absorption [19]. Considering that studies relating affective commitment and work engagement with GRHM practices are scarce, the first goal of this research is to understand the influence of GHRM practices on affective commitment and work engagement.

Even though sustainability is a major responsibility for organizations, individuals should also be accounted for in such a movement, as they decide daily to act more (or less) in an environmental manner [20]. Gardner and Stern [20] stated that individuals' behaviors are a major part of environmental problems, yet they are also their solutions. Thus, environmental issues are deeply connected to human values [21]. In line with Schwartz [22], human values can be defined as a desirable trans-situational goal varying in importance and serving as a leading principle in one's life. Based on this assumption, four values have been more associated with the individuals' environmental beliefs and behaviors [23–25]: biospheric (concern for the environment itself without interference from humans beings), altruistic (concern for the well-being and fair treatment for other human beings), egoistic (continuous assessment of cost and benefits that a choice has on one's personal resources), and hedonic values (focus on attaining pleasure and comfort and reducing effort). As a leading principle, individuals who prioritize the environment over subjects such as other human beings, one's life, or pleasure, will tend to feel more attached and connected to organizations that follow similar principles. Hence individual values are likely to deeply influence the way employees perceive the role of their organization regarding sustainability, in general, and green issues, in particular.

Although organizations through their HRM functions are key to developing a green consciousness in their employees, the role of human values, specifically the ones that are most related to the individuals' environmental beliefs and behaviors, are critical to support or resist such higher-level ambitions. This tri-dimensional view of the problem has not been previously addressed in research, which represents an important gap in existing knowledge. In fact, assuming that corporations through their HRM departments and practices are the only factors shaping employees' ecological concerns is a rather limited view of human nature. Human beings differ in the way they look at realities, which in the

current case means that while some people value green issues, some other people do not value such issues at all. Hence, the second goal of the current investigation is to understand the moderating effect of human values in the relationship between GHRM practices and affective commitment and work engagement.

The study of these relationships contributes to theory and practice in several ways. Firstly, there is a lack of studies about human values that are more related to environmental beliefs and behaviors; this is an increasingly relevant problem, due to the global urgency to deal with the environment and sustainable risks. This research introduces the topic of human values, and how they influence final behaviors in organizations. Secondly, organizational outcomes such as affective commitment and work engagement are central to HRM, in general, and GHRM, in particular, but to date, no studies have addressed the intervening role of human values in such relationships. The current work aims to contribute to theory by understanding the complex interactions between GRHM, human values, affective commitment, and work engagement. At a managerial level, this research aims to deepen the understanding of GHR practices, including how to measure them, and to understand how human values interact with critical employee outcomes.

## 2. Literature Review

### 2.1. GHRM Practices

In the last decade, research has identified sustainability as a practice capable of generating organizational value and attracting and retaining stakeholders [3]. Sustainability is perceived by organizations as a strategic management practice, consequently, it will be necessary for organizations to implement a short-, medium- and long-term plan integrating both internal and external influences and pressures [26]. This sustainability-oriented way of managing seeks to respond to new and old organizational challenges without compromising future generations in the access to primary goods [1]. The decision to support the business model in decisions that guarantee a sustainable future (for the world, for the organization itself, and for the employees) must be conscious and result from an organizational change process that considers all areas and departments and is led by the organization's managers [27]. The process of sustainable development is closely related to organizational performance, which triggers changes at the hierarchical, strategic, and operational design levels [1].

The Triple Bottom Line (TBL) approach emphasizes the three main pillars of sustainability: economic, political, and environmental [2–6]. The TBL means that organizations are starting to adjust their different areas/departments to guarantee the fulfillment of environmental goals. Environmental management practices may reduce more than 50% of the pollution levels which is possible with the introduction of management practices such as the development of new products or/and processes, the adoption of new and environmentally orientated technologies, and the implementation of environmental management programs [28–30]. Such initiatives may help organizations to minimize, for instance, the organization's and their employees' carbon footprints.

The HR departments are no exception. HRM practices and the HRM function aim at boosting employees' capabilities, performance, and well-being through which the organization can maintain (and increase) its competitive advantage [31]. When organizations invest in HRM systems, employees' competencies are expected to leverage the execution of strategic goals [32]. Following Paauwe and Boselie [33], it is also expected that HRM systems are aligned with organizational strategy, so that a positive impact on organizational performance is attained. Therefore, if organizations are willing to adjust their strategy to become more sustainable, the HRM system must walk a similar path by becoming more green-based.

This approach to HRM is called Green HRM (GHRM), and it emphasizes the activities developed by the HRM system to increase employees' environmental-supporting behaviors [34]. Ren and colleagues [16] defined GHRM as a phenomenon relevant to understanding the relationships between organizational activities that impact the natural

environment and the design, evolution, implementation, and influence of HRM systems. Tanova and Bayighomog [35] performed a systematic literature review on GHRM in service industries and concluded that although different authors have highlighted different facets of the HRM system, some frequently found dimensions include green recruitment and selection, green training, green performance management, green pay and reward, green involvement, green analysis, and job descriptions and empowerment. The authors also highlighted the increasing number of measures for GHRM practices, which tend to assess each practice separately [36,37], or employees' perceptions of the overall GHRM [38,39]. This trend looking at the relevance of HRM to green issues is still in its infancy, therefore research needs to address the extent to which HRM practices addressing sustainability and ecological issues have an impact on traditional individual outcomes, such as organizational commitment and work engagement. This is an important concern, as it expands the role of the HR function beyond the organizational boundaries, to include novel stakeholders, such as society and the planet.

The current study adopts the work of Cesário and colleagues [13], which proposed a five-dimension scale already adapted for the Portuguese population: Green Recruitment and Onboarding, Green Training, Green Performance Management and Rewards, Green Internal Communication, and Green Sustainable Culture. Considering the research goals and the approach of Dumont and colleagues [39] and Kim and colleagues [38], which considered the advantages of using a one-dimensional measure of GHRM, this research adapted the proposed scale using one item from each dimension covering all dimensions of the GHRM practices.

### 2.2. Affective Commitment

Commitment has been defined in several ways, however, there seems to be some level of consensus around Meyer and Allen's theoretical framework, including their assertion regarding the relevance of commitment to productivity, competitiveness, and well-being [40,41]. Commitment can be defined as a bond that characterizes the employee's relationship with the organization, which means it influences the strength of membership to a company [42]. The emotional side of this force, called affective commitment, has gained a special interest in the literature. Affective commitment is the employees' emotional bond to their organization, and it has been said to predict dedication and loyalty through a high sense of belonging and identification. Studies have confirmed a strong negative relationship between affective commitment and employee retention, and a strong and positive relationship with performance [17,43].

As an outcome, previous studies have shown a positive and significant relationship between HR practices and affective commitment. Yang [44] studied the mediating role of affective commitment in the relationship between high-involvement HR practices and organizational citizenship behaviors. The results confirmed the mediating role of affective commitment and the strong and positive relationship between high-involvement HR practices and affective commitment. More recently Alqudah, Carballo-Penela, and Ruzo-Sanmartín [45] found a positive relationship between affective commitment and communication, clear job description, extensive training, and participation. Results-oriented appraisal and selective staffing were not significantly related to affective commitment in the same study.

Bal and colleagues [46] investigated both developmental HRM and accommodative HRM, and their effects on commitment and work engagement. The results showed that developmental HRM relates to higher levels of commitment and engagement through the mediating role of the psychological contract. Therefore, higher levels of relational psychological contracts are associated with higher levels of commitment and engagement. As for the accommodative HRM, the results showed an increasing commitment for people endorsing selection and compensation strategies. Finally, in 2019 Aboramadan and colleagues [47] confirmed an important association between HRM and organizational commitment, which further reinforces the relationship between HR practices and commit-

ment. Research investigating such association with a HR function oriented towards green issues is much scarcer. The novelty of the topic may explain such a gap, however, authors investigating the ecological dimension of HR have found limited support for the view that green-oriented HRM practices influence employees' affective commitment [48–50]. These works suggest that:

**Hypothesis 1 (H1).** *GHRM practices are positively related to affective commitment.*

### 2.3. Work Engagement

Work engagement is defined as a positive, fulfilling, and work-related psychological state that encompasses three main dimensions: vigor (high standards of mental resilience and energy as well as willingness to invest effort in work), dedication (which portrays emotions such as inspiration, pride, and challenge), and absorption (concentration and happiness when performing tasks and the feeling that time passes quickly) [19].

Following the same pattern as affective commitment, research has shown a positive link between HR practices and work engagement [51–53]. For instance, the study of Aktar [54] focused on the relationship between HRM and work engagement and the mediating role of organizational commitment. The findings suggest a positive relationship and a partial mediating role of commitment. Thus, based on previous research, when employers implement successful HR practices, employees tend to be more engaged in their work and this will lead to better outcomes and positive attitudes [55,56].

Research on the relationship between GHRM practices and work engagement has been scarce since most of the work related to engagement has been focused on green engagement [57–59]. However, recent studies have already established a positive relationship between GHRM practices and work engagement. For instance, Darban, Karatepe, and Rezapouraghdam [60] tested the mediating role of work engagement in the relationship between GHRM practices and absenteeism and green recovery performance, and the results suggested that a positive perception of GHRM practices boosts work engagement. Karatepe, Hsieh, and Aboramadan's [61] study also showed a positive relationship between GHRM practices and work engagement. Therefore, the following hypothesis can be put forward:

**Hypothesis 2 (H2).** *GHRM practices are positively related to work engagement.*

### 2.4. Environmental Personal Values

Personal values can be defined as concepts or beliefs about behaviors or desired states that transcend specific situations, guide, select and/or evaluate behavior and events, and are ordered by their relative importance [62]. Personal values can be classified as cognitive representations of three universal human needs: biological, social, and institutional; and they express individual, collective or mixed interests [63]. Schwartz's theory is one of the most recognized in the literature, covering social and personality psychology, including a measurement instrument that has been tested in different cultural contexts [24].

Schwartz [63] characterized values as beliefs with a motivational source; in that sense, they relate to desirable ends and forms of behaviors, transcend specific situations and actions, are used as criteria for evaluation, and can be ordered by their relative importance concerning other values to form a system of priorities. As such, values are seen as desirable, trans-situational goals that vary in importance and serve as life principles for a person or other social entity [24]. Schwartz proposed 10 motivational types of values, integrated into a two-dimensional space encompassing four separate value clusters. The first dimension is openness to change versus conservatism, and differentiates values that emphasize self-direction and independence in opposition to tradition and conformity. The second dimension, self-transcendence versus self-enhancement, distinguishes values such as universalism and benevolence from those that pursue personal goals or self-enhancement,

like power and achievement. De Groot and Steg [24] highlighted that individuals with a prosocial (high self-transcendence) value orientation tend to maximize outcomes for others. In contrast, the proselves individuals (high self-enhancement) focus on themselves instead of others.

Studies on individuals' environmental attitudes and behaviors have included human values orientations. Therefore, based on the Schwartz approach, results have shown that individuals with a prosocial/self-transcendence value orientation tend to interact positively with environmental behaviors [64,65] and negatively with proselves/self-enhancement [24,66].

More recently, some authors have proposed a value orientation distinction focused on the environment [24,25,66,67]. These and other authors state four values most significant in predicting environmental beliefs and behaviors: two related to self-transcendence and two related to self-enhancement. The values related to self-transcendence—biospheric and altruistic—focus on others and the environment. In particular, the biospheric value is entirely focused on the environment and has no link with the individuals. Thus, individuals who base their actions on biospheric values tend to make decisions based on the perceived outcome for the environment and the ecosystem [68]. The altruistic value is based on the assumption that individuals' actions may affect the welfare of others; thus, today's decisions may be based on the possible consequences for future generations [69].

On the other hand, based on the self-enhancement dimension, two other environmental values are proposed: egoistic and hedonic. The egoistic value is characterized by the individual assessment of the cost and benefits of a certain (environmental) action for him/herself. In this sense, it is also based on individual personal power and resources to be used for self-concern [25], whereas the hedonic value is mainly related to the feeling of comfort and pleasure in completing a certain (environmental) action with a minimum effort [70].

Previous research has been showing that self-transcendence values (biospheric and altruistic) are positively related to environmental-friendly behaviors, whereas self-enhancement values (egoistic and hedonic) are negatively related [24,25].

Research on the buffering effect of personal environmental values on the relationship between GHRM practices and individual outcomes is still relatively scarce. More specifically, the linkages between personal environmental values, GHRM, commitment, and engagement have not been addressed in the literature. Some research [51] has examined cultural orientations, such as collectivism and power distance orientation, as moderators on the links between high-performance HR practices and perceived organizational support. Such research suggests that cultural values act as critical contextual factors, such that the relationship between HR practices and perceived organizational support, is stronger when collectivism is high and when power distance orientation is low. Another study by Zúñiga, Aguado, and Cabrera-Tenecela [71] analyzed the protestant work ethics in the relationship between HR practices and work engagement and organizational citizenship behavior, and identified that the dimensions of protestant work ethics—leisure and centrality of work, are moderators between HR practices and work engagement.

Based on these works the following hypotheses are outlined and the conceptual model is presented (Figure 1):

**Hypothesis 3.1 (H3.1).** *Biospheric value moderates the relationship between GHRM practices and affective commitment: the relationship between HRM and affective commitment is stronger when biospheric values score higher.*

**Hypothesis 3.2 (H3.2).** *Biospheric value moderates the relationship between GHRM practices and work engagement: the relationship between HRM and work engagement is stronger when biospheric values score higher.*

**Hypothesis 4.1 (H4.1).** *Altruistic value moderates the relationship between GHRM practices and affective commitment: the relationship between HRM and affective commitment is stronger when altruistic values score higher.*

**Hypothesis 4.2 (H4.2).** *Altruistic value moderates the relationship between GHRM practices and work engagement: the relationship between HRM and work engagement is stronger when altruistic values score higher.*

**Hypothesis 5.1 (H5.1).** *Egoistic value moderates the relationship between GHRM practices and affective commitment: the relationship between HRM and affective commitment is weaker when egoistic values score higher.*

**Hypothesis 5.2 (H5.2).** *Egoistic value moderates the relationship between GHRM practices and work engagement: the relationship between HRM and work engagement is weaker when egoistic values score higher.*

**Hypothesis 6.1 (H6.1).** *Hedonic value moderates the relationship between GHRM practices and affective commitment: the relationship between HRM and affective commitment is weaker when hedonic values score higher.*

**Hypothesis 6.2 (H6.2).** *Hedonic value moderates the relationship between GHRM practices and work engagement: the relationship between HRM and work engagement is weaker when hedonic values score higher.*

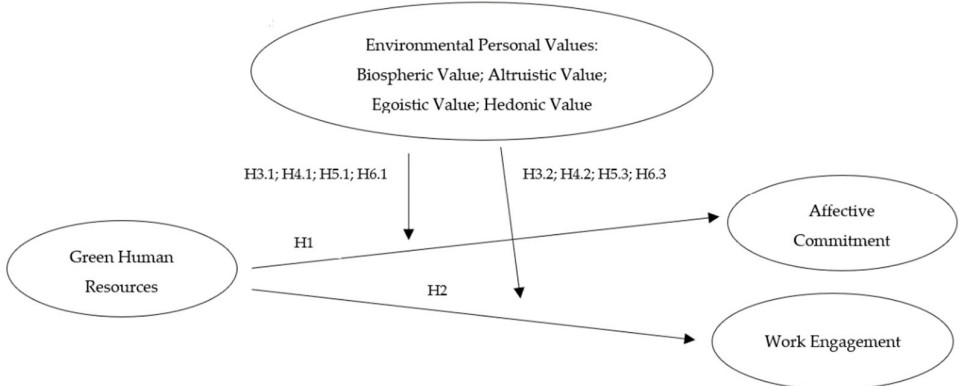

**Figure 1.** The Research Conceptual Model.

### 3. Method

*3.1. Data Collection Procedure*

This study had the voluntary participation of 532 professionals working in organizations in Portugal. Before the data collection, the selected scales were translated into Portuguese. The translation was carried out by two researchers who possessed experience in organizational behavior and were fluent in English. The questionnaire was given to 5 students to check face validity. The survey was created on Qualtrics, and the link was sent to contacts of the researchers via email or social media. The data collection process was non-probabilistic, convenience, intentional, and snowball-type. The data was collected in March 2021. All the participants were over 18 years old and employed at the moment of their participation in the study. At the beginning of the questionnaire, the research goals were explained. The survey contained information about the study's goals, and confidentiality and anonymity were guaranteed.

The questions comprised sociodemographic data (i.e., age, gender, tenure, sector), three scales measuring GHRM practices, organizational commitment, and work engage-

ment, and a final part assessing value orientations (hedonic, egoistic, altruistic, and biospheric values).

Concerning common method biases, in this research, recommendations by Podsakoff, MacKenzie, Lee, and Podsakoff [72] were used. These included informing participants of the required procedures to ensure confidentiality, emphasizing that there were no right or wrong answers, presenting the survey by instrument, and organizing the items in a random manner.

### 3.2. Data Analysis Procedure

To test the instrument's validity and the factor structure, exploratory and confirmatory factor analyses were performed with SPSS Statistics 27 and AMOS 27. Model fit was assessed via chi-square ($\chi^2$) $\leq 5$; for the goodness-of-fit index (GFI) > 0.90; for the comparative fit index (CFI) > 0.90; for root mean square error the approximation (RMSEA) $\leq 0.08$. Finally, the internal consistency of each instrument was tested with Cronbach's alpha ($\alpha > 0.7$).

The next step was to compute descriptive statistics (mean and standard deviation) and Pearson's correlations to analyze the associations between the variables under study. For a more robust analysis of the influence of a predictor variable (i.e., GRHM) on two criterion variables (i.e., affective commitment and work engagement; Hypotheses 1 and 2), a linear regression analysis was performed. The remaining hypotheses were tested via PROCESS 3.5 macro, model 1 [73]. SPSS's PROCESS 3.5 macro was used because it is a robust logistic regression path analysis modeling tool widely used in social, psychological, business, and health sciences. The moderating hypotheses were tested with eight different models. A bootstrap of 10,000 and bias-corrected 95 percent bootstrap confidence intervals for the conditional indirect effects were used. In the command of PROCESS, it was also asked to center the variables.

### 3.3. Sample

A total of 532 participants filled in the questionnaire, 63.7% of which were female and 36.1% male. Regarding age groups, 31% of the participants were in the 18–25 year-old group, 29.5% in the 26–33 year-old, 13.3% in the 34–41 year-old, and 13.9% in the 42–49 group. Concerning academic level, 39.7% completed secondary education, and 34.6% have a bachelor's degree. About 25% of participants have a post-graduate or higher degree. 51.7% of participants have a tenure higher than three years and 20.5% were in their organizations for one year.

### 3.4. Instruments

Green Human Resources Management Practices (GHRM practices). The concept of GHRM practices was measured with 6 items from Cesário and colleagues [13]. Two examples of items are "In the selection process of new employees, my company values environmentally conscious candidates", and "My company is committed to promoting an organizational culture oriented towards environmental sustainability". A five-point Likert-type response scale was used, ranging from 1 (totally disagree), to 5 (totally agree).

After performing an exploratory factor analysis, a KMO of 0.91 was obtained with a unidimensional solution which explained a total variance of 73.1%. The confirmatory factor analysis revealed good fit indices: $\chi^2/df = 1.5$; GFI = 0.96; CFI = 0.99; RMSEA = 0.031. Factor loadings vary from 0.70 to 0.89. The scale also registered a good internal consistency: Cronbach alpha = 0.92.

#### 3.4.1. Affective Commitment

Affective commitment was captured using Turker's [74] nine-item scale, which is a shorter version of the 15-item Organizational Commitment Questionnaire [18]. Sample items: "*I am willing to put in a great deal of effort beyond that normally expected in order to help this organization be successful*", and "*For me, this is the best of all possible organizations for which*

*to work*". A five-point Likert-type response scale was used, ranging from 1 (totally disagree), to 5 (totally agree).

After performing the exploratory factor analysis, a KMO of 0.90 was obtained with a unidimensional solution which explained a variance of 55.7%. The results from confirmatory factor analysis also showed a good fit with the data: $\chi^2/\text{df} = 3.5$; GFI = 0.96; CFI = 0.97; RMSEA = 0.069. Factor loadings vary from 0.51 to 0.85. As for internal consistency, the affective commitment scale presented a Cronbach's alpha of 0.89.

### 3.4.2. Work Engagement

The UWES (Utrecht Work Engagement Scale) [75] was used to measure work engagement. UWES is a three-dimensional scale, however, in the current case, a one-dimensional solution was followed because the aim of the current work was to focus on the moderating role of personal environmental values on the relationship between GHRM and affective commitment and work engagement as a positive, fulfilling and work-related psychological state and not the influence of the predictor and moderators on work engagement's specific dimensions. Sample items include "*At my work, I feel bursting with energy*", and "*At my job, I am very resilient, mentally*". As in the previous cases, a five-point Likert-type response scale ranging from 1 to 5 was used.

After performing the one-factor exploratory factor analysis, a KMO of 0.94 was obtained with a unidimensional solution which explained 47.9% of the final total variance. Regarding the confirmatory factor analysis, the goodness-of-fit indices are acceptable: $\chi^2/\text{df} = 7.4$; GFI = 0.85; CFI = 0.84; RMSEA = 0.010. Factor loadings vary from 0.46 to 0.84. The scale internal consistency showed a Cronbach's alpha of 0.92.

### 3.4.3. Environmental Value Orientations

The Environmental Portrait Value Questionnaire (E-PVQ) [25] was used to measure value orientations. The scale is composed of 16 items grouped in 4 dimensions (values). The hedonic dimension was measured with three items (e.g., "It is important to [him/her] to have fun"); the egoistic dimension has five items (e.g., "It is important to [him/her] to have money and possession"); the altruistic dimension has five items (e.g., "It is important to [him/her] that there is no war or conflict"; and the biospheric dimension has four items (e.g., "It is important to [him/her] to be in unity with nature"). A five-point Likert-type response scale, ranging from 1 to 5 was used.

After performing an exploratory factor analysis, a KMO of 0.88 was obtained with a multidimensional solution which explained a variance of 47.9%. The results from the confirmatory factor analysis showed a high goodness-of-fit: $\chi^2/\text{df} = 3.3$; GFI = 0.92; CFI = 0.92; RMSEA = 0.062; Factor loadings vary from 0.48 to 0.81. As for internal consistency, the hedonic scale registered a Cronbach's alpha of 0.71, the egoistic 0.74, the altruistic 0.80, and the biospheric 0.85.

The common method biases [72] were addressed via Harman's single-factor test through an exploratory factor analysis where all items were aggregated to a single factor. The results show that the common variance of all items in the survey is greater than the benchmark value of 40.9% established by Podsakoff and colleagues [71] for attitudinal measures. If the value of the variance extracted by the common factor is lower than the reference value, one can infer that there is no major impact on the estimation of the proposed model. Moreover, results show an extracted variance of 26% which is lower than the cut-off point put forward by the same authors.

## 4. Results

### 4.1. Descriptive Analysis

The means and standard deviations of the study variables are presented in Table 1.

**Table 1.** Descriptive Statistics of the Main Variables.

|  | Mean | SD | 1 | 2 | 3 | 4 | 5 | 6 | 7 |
|---|---|---|---|---|---|---|---|---|---|
| 1. GHRM practices | 2.84 | 1.15 | 1 | | | | | | |
| 2. Altruistic value | 4.68 | 0.44 | 0.032 | 1 | | | | | |
| 3. Biospheric value | 4.65 | 0.47 | 0.015 | 0.662 ** | 1 | | | | |
| 4. Hedonic value | 4.54 | 0.49 | 0.026 | 0.484 ** | 0.385 ** | 1 | | | |
| 5. Egoistic value | 3.04 | 0.68 | 0.133 ** | −0.102 ** | −0.049 | 0.142 ** | 1 | | |
| 6. Affective Commitment | 3.61 | 0.72 | 0.475 ** | 0.203 ** | 0.090 * | 0.168 ** | 0.250 ** | 1 | |
| 7. Work Engagement | 3.54 | 0.67 | 0.343 ** | 0.193 ** | 0.113 ** | 0.167 ** | 0.208 ** | 0.727 ** | 1 |

** $p < 0.01$; * $p < 0.05$.

The results show that GHRM practices in the participant organizations are not yet well established (M = 2.84, SD = 1.15). As described above, the scale asked about the use of several HRM practices encouraging sustainability and green practices, from culture to selection, and from reward strategies to engagement in green activities. Neither the items of the scale nor the aggregated index registered answers above 4 (scale 1–5).

Concerning the outcome variables—affective commitment and work engagement—participants evaluated slightly above the central point (M = 3.61, SD = 0.72; M = 3.54, SD = 0.67, respectively). In addition, the results also suggest that participants evaluated in a similar way both affective commitment and work engagement.

Finally, regarding individuals' environmental beliefs and behaviors, the results suggested high levels of altruistic, biospheric, and hedonic values (M = 4.68, SD = 0.44; M = 4.65, SD = 0.47; M = 4.54, SD = 0.49, respectively), and less on the egoistic value (M = 3.04, SD = 0.68).

This initial analysis also showed that only the egoistic value correlates (positively) with the GHRM practices (r = 0.133, $p < 0.01$), which is an unexpected result. None of the other values correlated with GHRM practices. The results also suggest that all environmental values are related to affective commitment, with the egoistic value showing the highest score (altruistic value: r = 0.203, $p < 0.01$; biospheric value: r = 0.09, $p < 0.05$; hedonic value: r = 0.168, $p < 0.01$; egoistic value: r = 0.250, $p < 0.01$). The same pattern is found with work engagement (altruistic value: r = 0.193, $p < 0.01$; biospheric value: r = 0.113, $p < 0.01$; hedonic value: r = 0.167, $p < 0.01$; egoistic value: r = 0.208, $p < 0.01$).

In line with the hypotheses, these results suggest a significant and positive correlation between GHRM practices and affective commitment and work engagement (r = 0.475, $p < 0.01$; r = 0.343, $p < 0.01$, respectively). Finally, in line with previous research (Cesário and Chambel, 2017), the correlation between affective commitment and work engagement is significant and positive (r = 0.727, $p < 0.01$).

*4.2. Hypothesis Testing*

To test the relationship between GHRM practices and affective commitment, results of the linear regression showed that GRHM practices positively influence affective commitment (β = 0.475; $p < 0.01$; $R^2$ = 0.226).

For work engagement, the results present a similar pattern, although a smaller variance was registered (work engagement: β = 0.343; $p < 0.01$; $R^2$ = 0.118). In sum, all these results point to the confirmation of both H1 and H2. Regarding the moderating role of the four environmental values in the relationship between GRHM practices and affective commitment (Table 2), it suggested that only the biospheric value moderates this relationship (B = −0.11; SE = 0.05; t = −2.07; $p = 0.030$; LLCI =−0.22, ULCI =−0.00; Table 2).

**Table 2.** The Moderating Role of the Environmental Personal Values in the relationship between GHRM and Affective Commitment.

| Variable | B | SE | t | p | 95% CI |
|---|---|---|---|---|---|
| GHRM → Affective Commitment ($R^2 = 0.26$; $p < 0.05$) | | | | | |
| Constant | 3.61 | 0.02 | 132.6 | 0.000 | [3.56,3.66] |
| GHRM | 0.29 | 0.02 | 11.47 | 0.000 | [0.24,0.34] |
| Altruism | 0.29 | 0.05 | 5.59 | 0.000 | [0.19,0.40] |
| GHRM X Altruism | −0.03 | 0.05 | −0.63 | 0.524 | [−0.14,0.07] |
| GHRM → Affective Commitment ($R^2 = 0.24$; $p < 0.05$) | | | | | |
| Constant | 3.61 | 0.02 | 130.7 | 0.000 | [3.56,3.67] |
| GHRM | 0.30 | 0.02 | 11.51 | 0.000 | [0.25,0.35] |
| Biospheric | 0.10 | 0.05 | 1.88 | 0.06 | [−0.00,0.21] |
| GHRM X Biospheric | −0.11 | 0.05 | −2.07 | 0.030 | [−0.22,−0.00] |
| GHRM → Affective Commitment ($R^2 = 0.25$; $p < 0.05$) | | | | | |
| Constant | 3.61 | 0.02 | 131.9 | 0.000 | [3.56,3.66] |
| GHRM | 0.29 | 0.02 | 11.57 | 0.000 | [0.24,0.35] |
| Hedonic | 0.22 | 0.04 | 4.59 | 0.000 | [0.12,0.32] |
| GHRM X Hedonic | −0.05 | 0.05 | −1.04 | 0.296 | [−0.15,0.04] |
| GHRM → Affective Commitment ($R^2 = 0.26$; $p < 0.05$) | | | | | |
| Constant | 3.61 | 0.02 | 130.7 | 0.000 | [3.56,3.67] |
| GHRM | 0.28 | 0.02 | 10.98 | 0.000 | [0.23,0.33] |
| Egoistic | 0.20 | 0.04 | 4.64 | 0.000 | [0.11,0.28] |
| GHRM X Egoistic | −0.01 | 0.03 | −0.38 | 0.704 | [−0.08,0.05] |

As can be observed in the interaction plot (Figure 2), when participants recognize that their organizations develop GRHM practices, then their affective commitment is higher independently of their biospheric value. However, if the organization presents a low score on GRHM practices, then their affective commitment is lower than when the GRHM practices are more established. The moderating role of the biospheric value is clear when GHRM practices are lower, increasing the affective commitment for the participants that present higher levels of biospheric values. That said, only H3.1 is confirmed.

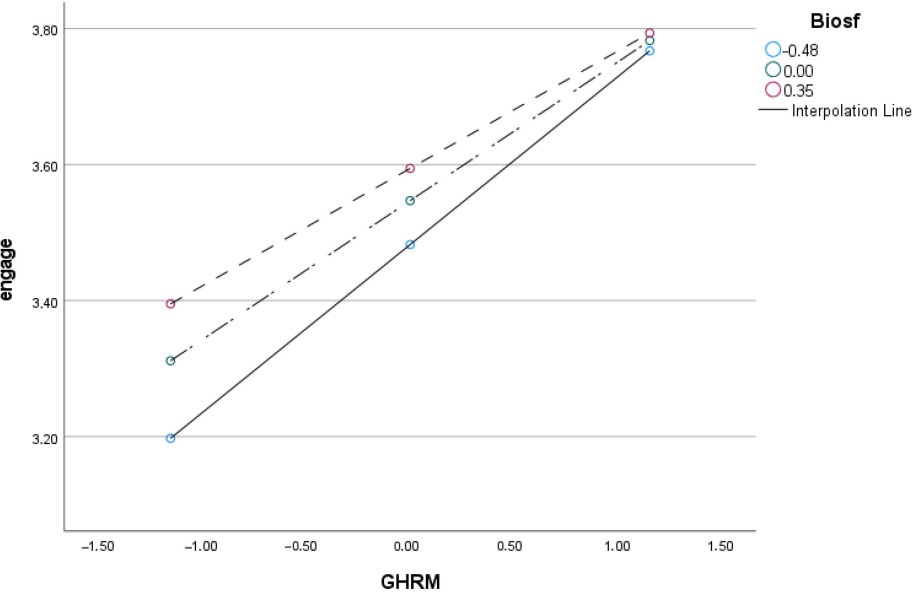

**Figure 2.** Interaction Plot GHRM X Biospheric to Affective Commitment.

For the moderating role of the environmental values in the relationship between GRHM practices and work engagement, the results are similar: only the biospheric value

moderates the aforementioned relationship (B = −0.09; SE = 0.04; t = −2.01; *p* = 0.044; LLCI = −0.17, ULCI = −0.00; Table 3). The interaction plot shows the same pattern as for affective commitment (Figure 3), hence the biospheric value assumes a more relevant role when GRHM practices are lower, increasing the work engagement for the participants who present higher levels of biospheric value. Only H3.2 is confirmed.

**Table 3.** The Moderating Role of the Environmental Personal Values in the Relationship Between GHRM and Work Engagement.

| Variable | B | SE | T | *p* | 95% CI |
|---|---|---|---|---|---|
| GHRM → Work Engagement ($R^2$ = 0.15; $p < 0.05$) | | | | | |
| Constant | 3.54 | 0.02 | 130.46 | 0.000 | [3.49,3.59] |
| GHRM | 0.19 | 0.02 | 7.45 | 0.000 | [0.14,0.24] |
| Altruism | 0.28 | 0.07 | 3.86 | 0.000 | [0.13,0.42] |
| GHRM X Altruism | 0.02 | 0.06 | 0.41 | 0.680 | [−0.09,0.14] |
| GHRM → Work Engagement ($R^2$ = 0.13; $p < 0.05$) | | | | | |
| Constant | 3.54 | 0.02 | 129.9 | 0.000 | [3.49,3.60] |
| GHRM | 0.20 | 0.02 | 7.86 | 0.000 | [0.15,0.25] |
| Biospheric | 0.13 | 0.05 | 4.44 | 0.014 | [−0.02,0.24] |
| GHRM X Biospheric | −0.09 | 0.04 | −2.01 | 0.044 | [−0.17,−0.00] |
| GHRM → Work Engagement ($R^2$ = 0.14; $p < 0.05$) | | | | | |
| Constant | 3.54 | 0.02 | 130.6 | 0.000 | [3.49,3.60] |
| GHRM | 0.20 | 0.02 | 7.81 | 0.000 | [0.14,0.25] |
| Hedonic | 0.21 | 0.05 | 3.66 | 0.000 | [0.09,0.32] |
| GHRM X Hedonic | −0.04 | 0.04 | −0.87 | 0.380 | [−0.13,0.05] |
| GHRM → Work Engagement ($R^2$ = 0.14; $p < 0.05$) | | | | | |
| Constant | 3.54 | 0.02 | 129.2 | 0.000 | [3.48,3.59] |
| GHRM | 0.18 | 0.02 | 7.06 | 0.000 | [0.13,0.23] |
| Egoistic | 0.15 | 0.04 | 3.43 | 0.000 | [0.06,0.24] |
| GHRM X Egoistic | 0.02 | 0.04 | 0.67 | 0.500 | [−0.05,0.11] |

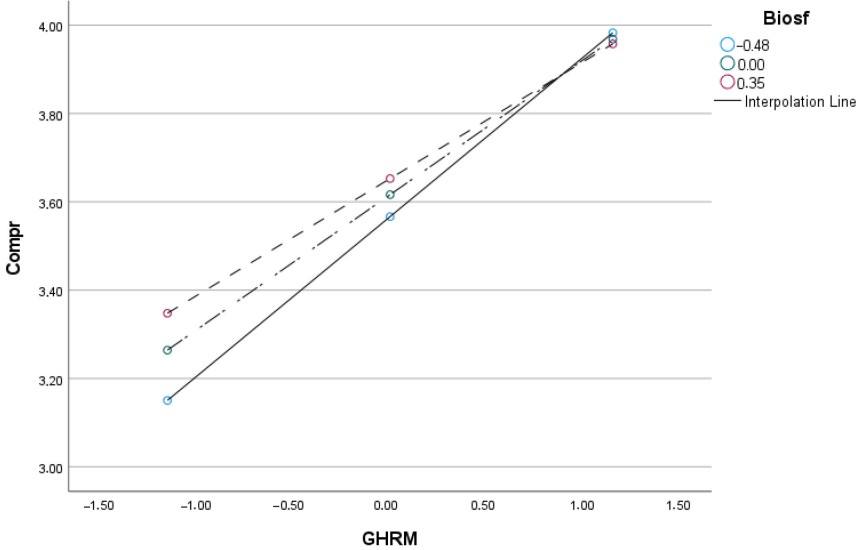

**Figure 3.** Interaction Plot GHRM X Biospheric to Work Engagement.

Following the hypothesis testing, Table 4 aggregates our final results.

**Table 4.** Research Hypothesis Results.

| Research Hypothesis | Conclusion |
| --- | --- |
| H1: GHRM practices are positively related to affective commitment | Confirmed |
| H2: GHRM practices are positively related to work engagement. | Confirmed |
| H3.1: Biospheric value moderates the relationship between GHRM practices and affective commitment: the relationship between HRM and affective commitment is stronger when biospheric values score higher | Confirmed |
| H3.2: Biospheric value moderates the relationship between GHRM practices and work engagement: the relationship between HRM and work engagement is stronger when biospheric values score higher | Confirmed |
| H4.1: Altruistic value moderates the relationship between GHRM practices and affective commitment: the relationship between HRM and affective commitment is stronger when altruistic values score higher | Not Confirmed |
| H4.2: Altruistic value moderates the relationship between GHRM practices and work engagement: the relationship between HRM and work engagement is stronger when altruistic values score higher | Not Confirmed |
| H5.1: Egoistic value moderates the relationship between GHRM practices and affective commitment: the relationship between HRM and affective commitment is weaker when egoistic values score higher | Not Confirmed |
| H5.2: Egoistic value moderates the relationship between GHRM practices and work engagement: the relationship between HRM and work engagement is weaker when egoistic values score higher | Not Confirmed |
| H6.1: Hedonic value moderates the relationship between GHRM practices and affective commitment: the relationship between HRM and affective commitment is weaker when hedonic values score higher | Confirmed |

## 5. Discussion and Conclusions

### 5.1. Discussion

The present research aimed to contribute to the UN 17 SDGs, in particular, SDG 8 "Decent Work and Economic Growth" and SDG 13 "Climate Change", as it focused on the environmental pillar of the TBL perspective. The goal was to explore the relationships between GHRM practices, affective commitment, work engagement, and employee environmental values (i.e., hedonic, egoistic, altruistic, and biospheric).

The first relevant result in the current research was the high average scores registered by the E-PVQ scale [25]. Individuals tend to emphasize the importance of several elements in their life and existence, from fun and joy, to money and material goods, and to the absence of war and the connection with the natural world. In sharp contrast with these high means, individuals tend to believe that corporations and HRM departments are doing little for the environment and other related topics, as shown by a lower mean score on the GHRM scale. The obtained results are in line with previous studies such as Dumont and colleagues [39], who also used a single measure for GRHM and registered means below the central point of the Likert scale. In Portugal, Cesário and colleagues [13] studied GRHM as a multidimensional construct and their findings suggested that each dimension was below or close to the central point of the Likert scale. The authors suggested that the most strategic GHRM practices (i.e., green sustainable culture) were better evaluated than the most operational ones (i.e., green recruitment and onboarding).

The high correlation between engagement and commitment is also noteworthy and it is in line with Cesário and Chambel's [17] work. Research confirms the association between these two outcomes [38], showing that when one feels connected to one's work, one also feels connected to one's organization. Which is a cause and which is a consequence, is still a matter of debate, as also found by Kim and colleagues [38].

Despite quite relatively low scores of GRHM in general, the confirmation of both hypotheses 1 and 2 show that individuals, regardless of their age group or educational level, tend to feel more engaged in their work and more committed to their organization when they perceive that HRM practices are supporting environmental concerns. These results confirm those of several authors [47,61], thus showing that the role of HRM is increasingly felt as encompassing not only the usual set of stakeholders (e.g., employees) but also stakeholders outside the borders of the organization, such as the environment and the planet. The impact of HRM practices on employees' behaviors, attitudes, and values, should be an obvious phenomenon, however, HRM has been traditionally focused on its role to foster and promote performance and organizational goals. In line with a rising

stream of studies, the current research indicates that employees tend to link HRM with their environmental concerns, which in turn suggests a wider role to the HRM function and its professionals. Whether or not the HRM function is ready to take on these broader responsibilities is something that should be at the core of discussions about the future of HRM.

The final set of results is related to the moderating roles of the four human values on the relationship between GRHM and organizational commitment (H3.1, H4.1, H5.1, and H6.1), on one hand, and, on the other hand, work engagement (H3.2, H4.2, H5.2, and H6.2). Of the proposed hypotheses, only H3.1 and H3.2 could be confirmed, i.e., the relationship between HRM and affective commitment is stronger when biospheric values score higher, and the relationship between HRM and work engagement is stronger when biospheric values score higher, respectively. No one of the other three values (altruistic, egoistic, and hedonic) revealed a moderating effect (Table 3).

Emphasizing biosphere means that individuals praise a life of harmony with nature and their surrounding natural environment. According to the results of this study, the association between GHRM practices and individual outcomes is reinforced when employees value the connection with nature and concerns about the environment. Values are powerful internal drivers of individual and collective decisions and actions [22], hence it is expected that they also affect human perceptions and interpretations of organizational behavior. When individuals value nature, it is likely that they tend to deliver more at work if they perceive that their organization shares the same concerns for the environment. Conversely, when they perceive that their organization has little or no concern for the environment, it is likely that they tend to underperform, regardless of the role of HRM in the company. Such perceptions are also likely to be affected by the employee's perception of GHRM, rather than just HRM.

Although the results suggested the moderating role of the biospheric value, such could not be confirmed as regards the altruistic value. As they are both self-transcendence values [24,25], it could be expected that individuals with high scores of altruism (which focuses on the welfare of others) could also moderate the relationship between GHRM and affective commitment and work engagement. This lack of impact of the altruistic value could be explained by its extreme focus on individuals, rather than on broader objects such as the environment.

Such results should be an eye-opener to HRM professionals, as more and more people of all ages are showing an interest in organizations' sustainable actions. As the HRM operates at the perceived boundaries between the organization and its employees, it is up to the HRM to show that the organization is emphasizing sustainability and responsible management toward caring for the environment. However, organizations should walk the talk, as doing the opposite of what is stated can result in a lack of trust towards managers and the organization, with negative consequences on commitment, engagement, and hence on productivity levels.

*5.2. Theoretical and Managerial Implications*

The findings suggest some theoretical and managerial implications. Regarding the theoretical implications, the current work contributes to the HRM and OB literature on the effects of GHRM practices on non-green individual outcomes, namely affective commitment, and work engagement. The results also show high scores of environmental personal values, which is in line with the growing global concern as expressed in the UN 17 SDGs. In this sense, this work contributes to a better understanding of the role of these specific human values in the relationship between organizational practices, through GHRM and attitudinal outcomes (i.e., affective commitment and work engagement). Concerning the moderating role of environmental personal values, the results suggest that although the altruistic, egoistic and hedonic values do not moderate these relationships, the biospheric value plays a crucial role by enhancing the effect of GHRM practices on affective commitment and work engagement. Another interesting result is related to the relatively low

scores of the GHRM practices variable. Respondents do not seem to be concerned or aware that the HRM function can play a role in the green movement. Thus, both theoretically and empirically this topic needs further development to increase knowledge of the implications of developing a HRM function based on sustainable goals. Another urgent matter is to understand how each HRM practice can become greener and how to evaluate their impact on financial outcomes, such as ROI (return on investment).

Based on these results and their theoretical implications, HRM scholars and practitioners need to realize that the success of managing people in organizations may depend strongly on human values, and that increasingly people are more alert to the way organizations look at the environment and at sustainability. HRM practices such as recruitment and selection can act as tools to attract and select candidates who have higher levels of biospheric values. Sustainability also needs to be part of the organizational strategy to guarantee a better fit across GHRM practices. If organizations do not work towards a sustainable strategy, it is likely that employees will not create positive perceptions of GHRM practices. Both internal and external communication are possible mechanisms to align the organization with its stakeholders. Should this trend continue at the actual pace, it is likely that employees will grow as a powerful change force inside the organization, which has to look less at short-term profits for some, and more at long-term profits for most.

### 5.3. Limitations and Future Research

Although this research has important strengths, certain limitations should be considered. The study has a cross-sectional design, therefore, causality cannot be established. Future studies using a longitudinal design would overcome this constraint. Although this research followed some of the recommendations by Podsakoff and colleagues [72], it is also undeniable that the use of self-report measures, the single source, and the lack of temporal separation when participants were answering the survey always raises common method bias concerns. Additionally, as a convenience sampling method was used, future studies could use a probabilistic sampling technique, to improve statistical power and representativity. In addition, all scales were translated by two researchers with experience in organizational behavior subjects and who were fluent in English. The final instrument was tested for its face validity with five people also fluent in English. Lastly, this research was conducted in Portugal, thus future research should replicate it in other countries to broaden and generalize the present results.

Because the aim of this research was to call attention to employee environmental values and their relationship with, and influence on other concepts (i.e., GHRM, affective commitment, and work engagement), the present study aimed to verify the influence of HR practices, as a unidimensional construct, on individual and non-green outcomes which were also unidimensional (i.e., affective commitment and work engagement). Future studies could use the multidimensional measure for GHRM practices [13] to look at each specific practice's influence on the studied outcomes. It could also be interesting to use the three dimensions of organizational commitment (adding continuance and normative commitment) and verify the influence on the three dimensions of work engagement (i.e., vigor, dedication, and absorption). In addition, and in line with Cahyadi and colleagues [76], it could also be interesting to add some organizational and green antecedents and outcomes such as green transformational leadership as a relevant antecedent, and organizational performance or employees' green behavior as outcomes.

**Author Contributions:** Conceptualization, J.F.S.G. and A.S.; methodology, V.A.; software, A.S.; validation, J.F.S.G.; formal analysis, A.S.; investigation, V.A.; resources, V.A.; data curation, A.S.; writing—original draft preparation, A.S.; writing—review and editing, J.F.S.G.; visualization, J.F.S.G.; supervision, J.F.S.G.; project administration, J.F.S.G. and A.S. All authors have read and agreed to the published version of the manuscript.

**Funding:** This research received no external funding.

**Institutional Review Board Statement:** Not applicable.

**Informed Consent Statement:** Informed consent was obtained from all subjects involved in the study.

**Data Availability Statement:** Not applicable.

**Acknowledgments:** The authors would like to thank the valuable comments received from reviewers and from the reviewer board of the journal. The authors also acknowledge the contribution of the participants who answered the survey.

**Conflicts of Interest:** The authors declare no conflict of interest.

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
