# Peer review of "The Effect of Green Human Resources Management Practices on Employees’ Affective Commitment and Work Engagement: The Moderating Role of Employees’ Biospheric Value"

_sustainability, doi:10.3390/su15032190_

Round 1
Reviewer 1 Report
Dear authors,
Thank you for sending me your article. Please find the enclosed file.
Regards,

Author Response
Dear Reviewer,
Thank you for the revision. Please find the enclosed file.
Best Regards,

Reviewer 2 Report
I would like to commend the authors for their efforts in this work. The paper is easy to follow and reasonably structured. However, I have some comments for its improvement:
1.) In the title, I believe that the word "and" in line 4 should perhaps be replaced by a comma (so that affective commitment and work engagement are both on the same level compared to Green HRM - and the suggested relationship refers to both together). At the same time, I feel that “the rise of the biospheric value” is a bit misleading. Biospheric value has been and will remain important for years and years. It has not risen now. Perhaps it would be more appropriate to change to e.g. “The importance of biospheric value as a moderator in the relationship between…”.
2.) In line 59 the authors state “reflected in different sectors from [e.g. 9]” – the sector they were meaning to mention is missing here (from which sector?).
3.) The paragraph that begins in line 161 does not belong in the literature review section. It belongs to the methodology section, as it explains the scale that was used in this study.
4.) In the beginning of the Method section (chapter 3), the authors should add a research model, where all their hypotheses will be visible to be reviewed in one figure. A similar figure should also be added in the discussion section (section 5), where the results of the statistical analyses will be visible and explain why/how the hypotheses have been confirmed or rejected.
5.) According to the methodology outlined, as I understand it, the translated questionnaires were not back-translated. This is an issue with methodology, as it may have affected the accuracy of the translation. Perhaps that is a limitation to mention in the limitations section.
6.) In line 313, you state that "All the participants were over 18 years old". Did you employ any other criteria to choose your sample? E.g. Were they all employed in a company etc.? Please clarify the selection criteria for your sample more thoroughly.
7.) In the paragraph after line 326, You need to provide some more details as per why you used each of the methods described. E.g. you used method A in order to test xxx. This method is appropriate according to X and Y researchers. In general you need to explain why you used each method (with the settings you used it), what you were trying to use it for, and why it is appropriate in this specific case. Also, although I am personally aware of the PROCESS tool, some of the readers may not be. Hence you need to explain what it is and how it works briefly.
8.) In the paragraph tha starts in line 365, you mention that you have treated UWES in its one-dimensional form. Why was a one-dimensional solution followed for UWES? Richer findings could be reported by also checking for the relationship with sub-scales. Please explain. Perhaps if you reported on the results of correlations with Vigor-Dedication-Absorption also, you would reach some additional findings? Please explain further or add accordingly.
9.) In the discussion section, the authors need to add a table where they recap (a) their hypotheses, (b) existing supporting evidence from the literature, (c) supporting evidence from the current study), and (d) Hypothesis confirmed / rejected (Yes or No). Each of the a-d can go on separate columns in the same table, so that the reader can clearly get a glimpse of the results of this study.
10.) In my view, Table 3, Figure 2, and their explanation (line 478 and on) belongs in the results section of the paper (and not in the discussion) – the same way that Table 2 and its explanation is already in the results section (both tables and figures refer to results).
11.) When the authors move the results to the results section (from the discussion) – based on the previous comment –, they will clearly see that the remaining discussion section will be quite short. Apart from its length, it also lacks content. The authors need to add in the discussion first the comparison of their results against the insight they have presented from the literature in the background section of the paper (compare your findings to what you have already noted as findings in existing literature). Apart from that they also need to shed more light into the theoretical and practical implications of their findings. I.e. they need to explain what it means that they reached these specific results, both for researchers and practitioners (businesses / HR departments) in specific. Finally, they need to also add some suggestions for research and practice, based on their findings, as well as insight they have found relative in the existing literature. Section 5.3 (theoretical and managerial implications) needs to also be moved up before the limitations and suggestions for future work) and enriched more. This is perhaps the most important part missing from this paper.
12.) As the whole paper is based only on questionnaire answers, there is an issue of single source bias that needs to be eliminated or at least reported as a limitation.
13.) The paper contains typos and needs to be proofread before finalization. Some of the typos I have found are: In line 438 "stablished" --> "established", Line 452 "fist" --> "first"
10.), Line 530 "biosferic" --> "biospheric". Please check thoroughly for additional.
Author Response
Firstly, we would like to thank you and the reviewers for taking the time and effort necessary to provide insightful guidance, which has contributed to improving this new version of the paper. We carefully considered the comments provided by Reviewer. Herein, we explain how we revised the manuscript based on those comments and recommendations.
Reviewer
Comment 1: The title does not convey the content of this research clearly. I believe that it would be more appropriate to change it to e.g. “The effect of Green Human Resources Management Practices on Employees’ Affective Commitment and Work Engagement: The moderating role of employees’ biospheric values”
We thank the reviewer for his/her insights and suggestions. Based on this and other reviewers’ suggestions, we decided to change the title of our paper to “Green Human Resources Management Practices, Affective Commitment, Work Engagement: The moderating role of employee environmental values”.
2.) In line 61 the authors state “reflected in different sectors from [e.g. 9]” – the sector they were meaning to mention is missing here (from which sector?).
Thank you for noticing. We added the missing information.
3.) The paragraph that begins in line 176 does not really belong in the literature review section. It belongs to the methodology section, as it explains the scale that was used in this study.
Although we understand this comment reviewer, our goal was to highlight that there are different theoretical approaches to GHRM, thus different instruments that focus on specific facets of the construct. By explaining this framework and clarifying which theoretical and methodological approach we followed in the literature review section it is our opinion that this information will be useful for the reader to understand our Hs.
4.) In the beginning of the Method section (chapter 3), the authors should add a research model, where all their hypotheses will be visible to be reviewed in one figure. A similar figure should also be added in the discussion section (section 5), where the results of the statistical analyses will be visible and explain why/how the hypotheses have been confirmed or rejected.
Thank you for this comment. We added the conceptual model in our paper. As for the figure presenting the statistical results, we added a table (table 3) with the confirmation (or not) of the Hs.
5.) According to the methodology outlined, as I understand it, the translated questionnaires were not back-translated. This is an issue with methodology, as it may have affected the accuracy of the translation. Perhaps that is a limitation to mention in the limitations section.
We added a note in the limitation section.
6.) In line 352, you state that "All the participants were over 18 years old". Did you employ any other criteria to choose your sample? E.g. Were they all employed in a company etc.? Please clarify the selection criteria for your sample more thoroughly.
In fact, being employed in the moment of the participation was also a criterion. We added the missing information.
7.) In the paragraph after line 373, you need to provide some more details as per why you used each of the methods described. E.g. you used method A in order to test xxx (e.g. why did you choose to use linear regression? When is it appropriate to employ it?). This method is appropriate according to X and Y researchers. In general you need to explain why you used each method (with the settings you used it), what you were trying to use it for, and why it is appropriate in this specific case. Also, although I am personally aware of the PROCESS tool, some of the readers may not be. Hence you need to explain what it is and how it works briefly.
We understand the reviewer suggestion. However, we believe that data analysis procedure clarifies all the steps necessary to analyse the data. We used standard information found in other papers.
8.) In the paragraph that starts in line 416, you mention that you have treated UWES in its one-dimensional form. Why was a one-dimensional solution followed for UWES? Richer findings could be reported by also checking for the relationship with sub-scales. Please explain. Perhaps if you reported on the results of correlations with Vigor-Dedication-Absorption also, you would reach some additional findings? Please explain further or add accordingly.
As a research team we discussed if we were to use work engagement as a uni or a multidimensional construct. In fact, such conversation included GHRM and commitment too.
Our research is essentially exploratory as far as the concept of employee environmental values is concerned. Hence we opted to keep simplicity with regards to the other concepts (namely GHRM, commitment and engagement), and leave for a later research the exploration of the full set of subdimensions. So our aim was to call attention to the employee environmental values and their relationship with, and influence in other more popular concepts. Figure 1 (new) shows the complexity of our model. Had we used the full multidimensionality of commitment and engagement, we would have worked with a model too complex for an exploratory research, and which would have probably required a larger sample.
9.) In the discussion section, the authors need to add a table where they recap (a) their hypotheses, (b) existing supporting evidence from the literature, (c) supporting evidence from the current study), and (d) Hypothesis confirmed / rejected (Yes or No). Each of the (a)-(d) mentioned above can go on separate columns in the same table, so that the reader can clearly get a glimpse of the results of this study.
10.) In my view, Table 3, Figure 2, and their explanation belongs in the results section of the paper (and not in the discussion) – the same way that Table 2 and its explanation is already in the results section (both tables and figures refer to results).
This answer addresses comments #9 and #10. We moved table 2 and figure 2 to the end of the results section, and we added table 3, to show which hypotheses were confirmed/not confirmed.
11.) When the authors move all the results to the results section (from the discussion – based on the previous comment), they will clearly see that the remaining discussion section will be quite short. Apart from its length, it also lacks content. The authors need to add in the discussion first the comparison of their results against the insight they have presented from the literature in the background section of the paper (compare your findings to what you have already noted as findings in existing literature). Section 5.3 (theoretical and managerial implications) needs to also be moved up before the limitations and suggestions for future work) and enriched more. This is perhaps the most important part missing from this paper. The authors need to shed more light into the theoretical and practical implications of their findings. I.e. they need to explain what it means that they reached these specific results, both for researchers and practitioners (businesses / HR departments) in specific – the current sub-section is quite brief and more exact suggestions need to be provided, based on existing literature. Finally, they need to also add some suggestions for research and practice, based on their findings, as well as insight they have found relative in the existing literature.
We understand the reviewer concern, as we moved up the theoretical and managerial implications and we added some suggestions for research and practice and added information regarding the theoretical implications. Regarding the discussion we added some new information and some studies that are (or not) in line with our results
12.) As the authors admit in their limitations, the whole paper is based only on questionnaire answers, and this creates an issue of single source bias. The authors need to explain why their research did not include further means of assessment, as well as suggest how this issue can be overcome in future research designs (what future researchers should do better/differently). They should accordingly suggest ways in which future research can corroborate their findings more accurately and/or strongly.
We are very grateful for this comment, as during the research team discussions, we debated lengthy the risk of common method bias and single source bias. Thus, we added: (1) The precautions taken during data collection (see new paragraph at the end of section 3.1); (2) a note regarding a EFA that was performed to verify if the extracted variance was lower than the recommended by Podsakoff and colleagues (2003) for attitudinal measure (new paragraph at the end of section 3.4); and, (3) a reference to CMB and Single Method Bias limitations (in section 5.2).
13.) The paper still contains numerous typos and needs to be thoroughly proofread before finalization. Some of the typos I have found are: In line 541 "stablished" --> "established", Line 573 "fist" --> "first", Line 716 "biosferic" --> "biospheric". Please check thoroughly for additional.
We thank the reviewer for noticing these. We amended them and gave the text a thorough final read.
In closing, we would like to thank the Editor for the opportunity to reformulate and send a new version of our manuscript and Reviewer again for their comments. We hope that we have dealt with the Reviewers’ suggestions satisfactorily and made all the adjustments requested by reviewers, both in form and substance.
Yours sincerely,
On behalf of my co-authors,
Reviewer 3 Report
Thank you for submitting the manuscript id sustainability-2157720 entitled “The rise of the biospheric value: The moderating role of environmental values on the relationship between Green Human Resources Management Practices and Affective Commitment and Work Engagement.” Here are my suggestions which could be helpful for improving it. (check attached pdf)
Good luck.

Author Response

(The authors gave the same response as above.)

Reviewer 4 Report
The article “The rise of the biospheric value: The moderating role of envi- ronmental values on the relationship between Green Human Resources Management Practices and Affective Commitment and Work Engagemen” takes up the very important theme of human resource management in a dynamic that is changing with unknown dynamics for humanity, which gives rise to specific impacts and associated challenges. One of these challenges is green human resource management (GHRM), which has been used to describe the management of people with a focus on ecological issues. In this perspective, an undoubted strength of the author's research presented in the article is the recognition that GHRM has a decisive impact on organisational performance, as it translates into all areas of the organisation. In this light, the planned strategic solutions in the area of GHRM should crucially constitute a coherent set of principles - shaping attitudes in organisations in a dynamically changing environment - which is determined by employees aware of shared values, characteristics and acting with a sense of unity.
It should be emphasised that further important and review-shaping premises occurred in the article, which are:
ü correctly formulated research hypotheses,
ü large research sample,
ü a statistically weak research sampling method (snowballing) with a very statistically sensitive analysis of the results obtained? This theme is explained by the authors of the article in the section on limitations and future research. It is a form of acknowledgement of the need to explore the problem under study through more correct research sampling - e.g. probabilistic sampling,
ü numerous threads that are not relevant to the research results (e.g. the issue of translation of scales - lines 305-316),
ü numerous results can be placed in tables achieving a clearer and comprehensible to the reader (e.g. lines 321-342, 343-374),
ü under testing, the data contained in the description duplicate those in Table 2 - perhaps it would be sufficient to add the references used to the result described? Similarly, in further parts of the article.
In recommending the article for publication, I propose to carry out the few - but important - corrections indicated above.
Author Response

(The authors gave the same response as above.)

Round 2
Reviewer 1 Report
Dear authors,
Thank you for sending me the revised article. Your article will be better and more interesting if it presents figures of research framework and structural model. Good luck!
Author Response
Firstly, we would like to thank you and the reviewers for taking the time and effort necessary to provide insightful guidance, which has contributed to improving this new version of the paper. We carefully considered the comments provided by Reviewer 1. Herein, we explain how we revised the manuscript based on those comments and recommendations.
Reviewer 1
Dear authors,
Thank you for sending me the revised article. Your article will be better and more interesting if it presents figures of research framework and structural model. Good luck!
Comment 1: We thank reviewer 1 for his/her suggestion. We added the conceptual model following the Hs presentations. Regarding the structural model, because we did not perform Structural Equations Models to test our Hs we decided to add a table that systemize our results.
In closing, we would like to thank the Editor for the opportunity to reformulate and send a new version of our manuscript and Reviewer1 again for their comments. We hope that we have dealt with the Reviewers’ suggestions satisfactorily and made all the adjustments requested by reviewers, both in form and substance.
Yours sincerely,
On behalf of my co-authors,
Reviewer 2 Report
I would like to commend the authors for their efforts in this work. The paper is easy to follow and reasonably structured. However, I have some comments for its improvement:
1.) The title does not convey the content of this research clearly. I believe that it would be more appropriate to change it to e.g. “The effect of Green Human Resources Management Practices on Employees’ Affective Commitment and Work Engagement: The moderating role of employees’ biospheric values”
2.) In line 61 the authors state “reflected in different sectors from [e.g. 9]” – the sector they were meaning to mention is missing here (from which sector?).
3.) The paragraph that begins in line 176 does not really belong in the literature review section. It belongs to the methodology section, as it explains the scale that was used in this study.
4.) In the beginning of the Method section (chapter 3), the authors should add a research model, where all their hypotheses will be visible to be reviewed in one figure. A similar figure should also be added in the discussion section (section 5), where the results of the statistical analyses will be visible and explain why/how the hypotheses have been confirmed or rejected.
5.) According to the methodology outlined, as I understand it, the translated questionnaires were not back-translated. This is an issue with methodology, as it may have affected the accuracy of the translation. Perhaps that is a limitation to mention in the limitations section.
6.) In line 352, you state that "All the participants were over 18 years old". Did you employ any other criteria to choose your sample? E.g. Were they all employed in a company etc.? Please clarify the selection criteria for your sample more thoroughly.
7.) In the paragraph after line 373, you need to provide some more details as per why you used each of the methods described. E.g. you used method A in order to test xxx (e.g. why did you choose to use linear regression? When is it appropriate to employ it?). This method is appropriate according to X and Y researchers. In general you need to explain why you used each method (with the settings you used it), what you were trying to use it for, and why it is appropriate in this specific case. Also, although I am personally aware of the PROCESS tool, some of the readers may not be. Hence you need to explain what it is and how it works briefly.
8.) In the paragraph that starts in line 416, you mention that you have treated UWES in its one-dimensional form. Why was a one-dimensional solution followed for UWES? Richer findings could be reported by also checking for the relationship with sub-scales. Please explain. Perhaps if you reported on the results of correlations with Vigor-Dedication-Absorption also, you would reach some additional findings? Please explain further or add accordingly.
9.) In the discussion section, the authors need to add a table where they recap (a) their hypotheses, (b) existing supporting evidence from the literature, (c) supporting evidence from the current study), and (d) Hypothesis confirmed / rejected (Yes or No). Each of the (a)-(d) mentioned above can go on separate columns in the same table, so that the reader can clearly get a glimpse of the results of this study.
10.) In my view, Table 3, Figure 2, and their explanation belongs in the results section of the paper (and not in the discussion) – the same way that Table 2 and its explanation is already in the results section (both tables and figures refer to results).
11.) When the authors move all the results to the results section (from the discussion – based on the previous comment), they will clearly see that the remaining discussion section will be quite short. Apart from its length, it also lacks content. The authors need to add in the discussion first the comparison of their results against the insight they have presented from the literature in the background section of the paper (compare your findings to what you have already noted as findings in existing literature). Section 5.3 (theoretical and managerial implications) needs to also be moved up before the limitations and suggestions for future work) and enriched more. This is perhaps the most important part missing from this paper. The authors need to shed more light into the theoretical and practical implications of their findings. I.e. they need to explain what it means that they reached these specific results, both for researchers and practitioners (businesses / HR departments) in specific – the current sub-section is quite brief and more exact suggestions need to be provided, based on existing literature. Finally, they need to also add some suggestions for research and practice, based on their findings, as well as insight they have found relative in the existing literature.
12.) As the authors admit in their limitations, the whole paper is based only on questionnaire answers, and this creates an issue of single source bias. The authors need to explain why their research did not include further means of assessment, as well as suggest how this issue can be overcome in future research designs (what future researchers should do better/differently). They should accordingly suggest ways in which future research can corroborate their findings more accurately and/or strongly.
13.) The paper still contains numerous typos and needs to be thoroughly proofread before finalization. Some of the typos I have found are: In line 541 "stablished" --> "established", Line 573 "fist" --> "first", Line 716 "biosferic" --> "biospheric". Please check thoroughly for additional.
Author Response

(The authors gave the same response as above.)

Reviewer 4 Report
I have no attention. I recommend the article for printing.
Author Response
Dear Reviewer,
We appreciate your feedback.
Best Wishes
Round 3
Reviewer 2 Report
I would like to commend the authors for their efforts in reviewing this paper. The revised version (in my view) has addressed the majority of my comments. However, since some of my comments have not been fully appreciated, I would like to turn the attention of the authors towards re-visiting and addressing them, as I believe that the effect of the paper will be quite positive.
Firstly, as per the title of the paper, although it has been changed, it still remains vague and does not reflect the content of the paper. If you compare it to the newly added research model (Figure 1), you will see that what is missing from the title is the description of what affects what in your study. Therefore, I strongly suggest that you add the missing words between Green HRM, Affective commitment and Work Engagement and utilize my originally suggested title: “The effect of Green Human Resources Management Practices on Employees’ Affective Commitment and Work Engagement: The moderating role of employees’ biospheric values”
My additional comments that have not been given as much attention as needed from the previous round follow (comments 7,8,9 copied and further commented on):
7.) In the paragraph after line 373, you need to provide some more details as per why you used each of the methods described. E.g. you used method A in order to test xxx (e.g. why did you choose to use linear regression? When is it appropriate to employ it?). This method is appropriate according to X and Y researchers. In general you need to explain why you used each method (with the settings you used it), what you were trying to use it for, and why it is appropriate in this specific case. Also, although I am personally aware of the PROCESS tool, some of the readers may not be. Hence you need to explain what it is and how it works briefly.
(Note Re 7: The authors suggest that they have followed the structure and content in other papers. I respect that, but firmly stand behind my original comment. There is a need to more clearly explain to the reader your reasoning behind the analysis methodology employed so that they are convinced that what you did is logical and appropriate for this specific case. Moreover, tools like the PROCESS tool are not universally known - although I personally am familiar with it, you need to explain what it is to an audience that may not have even heard of it before - it's not as popular as SPSS or other well-known packages).
8.) In the paragraph that starts in line 416, you mention that you have treated UWES in its one-dimensional form. Why was a one-dimensional solution followed for UWES? Richer findings could be reported by also checking for the relationship with sub-scales. Please explain. Perhaps if you reported on the results of correlations with Vigor-Dedication-Absorption also, you would reach some additional findings? Please explain further or add accordingly.
(Note Re 8: The authors have provided the rationale behind their choice in the response letter. I suggest that they also incorporate it within the paper's methodology also so that the readers will comprehend better).
9.) In the discussion section, the authors need to add a table where they recap (a) their hypotheses, (b) existing supporting evidence from the literature, (c) supporting evidence from the current study), and (d) Hypothesis confirmed / rejected (Yes or No). Each of the (a)-(d) mentioned above can go on separate columns in the same table, so that the reader can clearly get a glimpse of the results of this study.
(Note Re 9: Although a table has been added, there are 2 additional columns with supporting evidence missing - they need to be added in order for the table to be complete)
Author Response
Firstly, we would like to thank you and the reviewers for taking the time and effort necessary to provide insightful guidance, which has contributed to improving this new version of the paper. We carefully considered the comments provided by Reviewer. Herein, we explain how we revised the manuscript based on those comments and recommendations.
Reviewer 1
Comment 1: I would like to commend the authors for their efforts in reviewing this paper. The revised version (in my view) has addressed the majority of my comments. However, since some of my comments have not been fully appreciated, I would like to turn the attention of the authors towards re-visiting and addressing them, as I believe that the effect of the paper will be quite positive.
We thank the reviewer for his/her insights.
Firstly, as per the title of the paper, although it has been changed, it still remains vague and does not reflect the content of the paper. If you compare it to the newly added research model (Figure 1), you will see that what is missing from the title is the description of what affects what in your study. Therefore, I strongly suggest that you add the missing words between Green HRM, Affective commitment and Work Engagement and utilize my originally suggested title: “The effect of Green Human Resources Management Practices on Employees’ Affective Commitment and Work Engagement: The moderating role of employees’ biospheric values”
We agree with the reviewer, and we decided to change the title.
My additional comments that have not been given as much attention as needed from the previous round follow (comments 7,8,9 copied and further commented on):
7.) In the paragraph after line 373, you need to provide some more details as per why you used each of the methods described. E.g. you used method A in order to test xxx (e.g. why did you choose to use linear regression? When is it appropriate to employ it?). This method is appropriate according to X and Y researchers. In general you need to explain why you used each method (with the settings you used it), what you were trying to use it for, and why it is appropriate in this specific case. Also, although I am personally aware of the PROCESS tool, some of the readers may not be. Hence you need to explain what it is and how it works briefly.
(Note Re 7: The authors suggest that they have followed the structure and content in other papers. I respect that, but firmly stand behind my original comment. There is a need to more clearly explain to the reader your reasoning behind the analysis methodology employed so that they are convinced that what you did is logical and appropriate for this specific case. Moreover, tools like the PROCESS tool are not universally known - although I personally am familiar with it, you need to explain what it is to an audience that may not have even heard of it before - it's not as popular as SPSS or other well-known packages).
We added the missing information.
8.) In the paragraph that starts in line 416, you mention that you have treated UWES in its one-dimensional form. Why was a one-dimensional solution followed for UWES? Richer findings could be reported by also checking for the relationship with sub-scales. Please explain. Perhaps if you reported on the results of correlations with Vigor-Dedication-Absorption also, you would reach some additional findings? Please explain further or add accordingly.
(Note Re 8: The authors have provided the rationale behind their choice in the response letter. I suggest that they also incorporate it within the paper's methodology also so that the readers will comprehend better).
We added a note in the instruments section and also, we reinforce the need to study the WE’s specific dimensions on the limitations and future section.
9.) In the discussion section, the authors need to add a table where they recap (a) their hypotheses, (b) existing supporting evidence from the literature, (c) supporting evidence from the current study), and (d) Hypothesis confirmed / rejected (Yes or No). Each of the (a)-(d) mentioned above can go on separate columns in the same table, so that the reader can clearly get a glimpse of the results of this study.
(Note Re 9: Although a table has been added, there are 2 additional columns with supporting evidence missing - they need to be added in order for the table to be complete)
We understand the reviewer suggestion. After careful analysis and discussion with the research team, we decided to maintain the table as it is. Regarding column b) our research is essentially exploratory as far as the concept of employee environmental values is concerned. Hence our Hs were mainly build upon deductive approach on previous research related with similar constructs. Regarding column c) all results are presented both on the text and on table 1 and 2. He hope the reviewer understands our position.
In closing, we would like to thank the Editor for the opportunity to reformulate and send a new version of our manuscript and Reviewer again for their comments. We hope that we have dealt with the Reviewers’ suggestions satisfactorily and made all the adjustments requested by reviewers, both in form and substance.
Yours sincerely,
On behalf of my co-authors,